# Facile Fabrication of Diatomite-Supported ZIF-8 Composite for Solid-Phase Extraction of Benzodiazepines in Urine Samples Prior to High-Performance Liquid Chromatography

**DOI:** 10.3390/molecules26175209

**Published:** 2021-08-27

**Authors:** Li Du, Shaonan Xu, Han Wu, Tengwen Zhao, Xuesheng Wang, Manman Wang

**Affiliations:** 1School of Public Health, North China University of Science and Technology, Tangshan 063210, China; duli27@163.com (L.D.); xvshaonanxsn@163.com (S.X.); wuhan246@126.com (H.W.); ztwen_hbu@163.com (T.Z.); 2Hebei Province Key Laboratory of Occupational Health and Safety for Coal Industry, School of Public Health, North China University of Science and Technology, Caofeidian, Tangshan 063210, China

**Keywords:** benzodiazepines, diatomite, zeolitic imidazolate framework-8, solid-phase extraction, urine

## Abstract

A novel diatomite-supported zeolitic imidazolate framework-8 sorbent (ZIF-8@Dt-COOH) was in situ fabricated and developed for solid-phase extraction of three benzodiazepines (triazolam, midazolam and diazepam) in urine followed by high-performance liquid chromatography. ZIF-8@Dt-COOH was easily prepared by coating ZIF-8 on the surface of Dt-COOH and characterized by Fourier transform infrared spectra, X-ray powder diffractometry and scanning electron microscopy. Compared with bare Dt-COOH, the extraction efficiency of ZIF-8@Dt-COOH for the target was significantly increased from 20.1–39.0% to 100%. Main extraction parameters, including ionic strength and pH of solution, loading volume, washing solution, elution solvent and elution volume, were optimized in detail. Under optimum conditions, the developed method gave linearity of three BZDs in 2–500 ng/mL (*r* ≥ 0.9995). Limits of detection (*S/N* = 3), and limits of quantification (*S/N* = 10) were 0.3–0.4 ng/mL and 1.0–1.3 ng/mL, respectively. In addition, the average recoveries at three spiked levels (5, 10 and 20 ng/mL) varied from 80.0% to 98.7%, with the intra-day and inter-day precisions of 1.4–5.2% and 1.5–8.2%, respectively. The proposed method provided an effective purification performance and gave the enrichment factors of 24.0–29.6. The proposed method was successfully employed for the accurate and sensitive determination of benzodiazepines in urine.

## 1. Introduction

Benzodiazepines (BZDs, Figure 1), as a class of psychoactive drugs, are extensively prescribed in the therapy of insomnia, anxiety and convulsive attacks due to their hypnotic, anxiolytic, anticonvulsant and muscle-relaxant properties [1]. However, long-term use of BZDs leads to the risk of dependence, memory loss, fainting and cognitive disturbance [2,3]. Besides therapeutic usage, BZDs with alcohol and other drugs (sedatives, antidepressants and neuroleptics) may result in illicit drug abuse and even drug-facilitated assault cases and robbery [4,5]. Consequently, the accurate and sensitive monitoring of BZDs in pharmaceutical preparations, clinical or criminal examinations is of particular importance.

For biological analysis, sample pretreatment is an important process to isolate desired components and remove matrix interferences from complex matrices [6]. Numerous sample pretreatment techniques have been developed for the determination of BZDs in biofluids, such as liquid–liquid extraction [7], solid-phase microextraction (SPME) [8], magnetic solid-phase extraction (MSPE) [9] and solid-phase extraction (SPE) [10,11].

SPE is an attractive sample pretreatment methodology owing to the advantages of its high extraction efficiency, low solvent consumption and simplicity in operation [12,13]. Adsorbents in SPE, which play key roles in purification and isolation ability for trace analytes from complex matrices, have attracted increasing interest. Therefore, besides commercial SPE adsorbents C_8_, C_18_, etc., various novel carbon nanotubes, graphene, molecular imprinting polymers and covalent organic frameworks have been explored for sample pretreatment in food, biological, environmental and pharmaceutical analysis [14,15,16,17].

Metal–organic frameworks (MOFs) are an attractive coordination polymer constructed with metal ions and organic linkers by coordination bonds [18]. Their extraordinary properties, such as good thermal and mechanical stability, tunable pore topologies and high surface area, make MOFs attractive in the fields of separation, catalysis and sensing [19,20,21]. Additionally, the utilization of MOFs has attracted intensive attention in sample pretreatment with the mode of SPE, MSPE, dispersive solid-phase extraction (DSPE) and SPME [22,23,24,25]. Recently, ZIF-8@cellulose was prepared by simple hydrothermal synthesis and was used as an SPE adsorbent to analyze polycyclic aromatic hydrocarbons in the environment water. The developed method exhibited high sensitivity with a low limit of detection of 0.1–1.0 ng/L [22]. A novel Fe_3_O_4_-NH_2_@MOF-235 sorbent was proposed for the extraction of five benzoylurea insecticides from food and environmental samples, including honey, fruit juice and tap water [23]. ZIF-67 was synthesized and employed as a DSPE sorbent for monitoring of buprenorphine in biological fluids coupled with UHPLC-UV [24]. In addition, UiO-67-coated SPME fibers with good thermal stability were applied for the pretreatment of nitrobenzene compounds in environmental water samples before GC-MS [25].

Diatomite (Dt) composed of amorphous siliceous (SiO_2_·*n*H_2_O) is a kind of natural mineral, and exhibits ordered pore-size distribution and high porosity, as well as exceptional thermal, mechanical and chemical stabilities [26,27]. In addition, the abundance and availability of Dt have attracted particular interest due to its unique properties in the field of catalysis, filtration and adsorption [28,29,30]. For example, Dt was selected as an environmentally friendly support for the synthesis of Pd-M/Dt to catalyze hydrogenation of long-chain aliphatic esters [28]. A porous Dt ceramic was fabricated and used as an adsorbent for the removal of volatile organic compounds [29]. Additionally, an allophane/Dt nanocomposite with hierarchically porous structure was in situ synthesized for benzene removal [30].

Herein, diatomite-supported ZIF-8 (ZIF-8@Dt-COOH) combining the advantages of both Dt and ZIF was synthesized via simple in situ coating of ZIF-8 on the surface of Dt-COOH and applied as an SPE sorbent for isolation of BZDs in urine samples. ZIF-8, as a typical MOF material, is self-assembled from zinc nitrate hexahydrate metal ion and 2-methylimidazole (2-MeIM) ligands with a large surface area, and mechanical and chemical stability [31]. Recently, magnetic ZIF-8 sorbent, carbon and ZIF-8 composite-based membrane/adsorbent have been developed for the pretreatment of tetracycline antibiotics, caffeine and methamphetamine in water, caffeine beverages and urine samples [32,33,34]. In this work, a novel ZIF-8@Dt-COOH SPE sorbent was prepared and characterized via Fourier transform infrared spectra, X-ray powder diffractometry and scanning electron microscopy. The possible mechanism between ZIF-8@Dt-COOH and BZDs was discussed, and potential factors influencing the SPE process were also investigated in detail. The developed method of ZIF-8@Dt-COOH-based SPE coupled with HPLC was used for the determination of three BZDs in urine.

## 2. Results and Discussion

### 2.1. Synthesis and Characterization of ZIF-8@Dt-COOH

The ZIF-8 nanocrystals grown on Dt-COOH surface provide the interaction sites, and their density dominates the adsorption efficiency for the target compounds. In order to obtain ZIF-8@Dt-COOH with adequate dense and uniform morphology of ZIF-8, two parameters of Zn^2+^ concentration (0.04, 0.08 and 0.12 mmol/mL) and carboxylation cycle of Dt (1 and 2) were optimized. SEM images in Figure 2 show that when the concentration of Zn^2+^ is fixed at 0.04 mmol/mL, almost no growth of ZIF-8 crystals is found on the surface of both Dt-COOH(1×) and Dt-COOH(2×). As Zn^2+^ concentration reaches 0.08 mmol/mL, regular dodecahedron ZIF-8 nanocrystals with an average particle diameter of 300–400 nm are clearly observed on Dt-COOH(1×) and Dt-COOH(2×). The density of ZIF-8 particles alters with the increase in growth cycles of carboxylation (Figure 2a–f), and more ZIF-8 nanocrystals form on Dt-COOH(2×) microspheres than Dt-COOH(1×), which can be attributed to adequate carboxyl groups providing more affinity sites for the anchoring of Zn^2+^. Therefore, the concentration of Zn^2+^ was fixed at 0.08 mmol/mL, and the cycle of carboxylation of Dt was selected to be 2. In this work, the in situ growth of ZIF-8 on Dt-COOH(2×) was achieved with mechanical stirring in a 70°C water bath for 20 min, offering a convenient and fast synthesis.

ZIF-8@Dt-COOH(2×) was further characterized by FT-IR and XRD. Figure 3a shows the FT-IR spectra of ZIF-8, Dt-COOH(2×) and ZIF-8@Dt-COOH(2×). The main peaks appearing at 2926, 1600, 1145 and 995 cm^−1^ are caused by C-H, C=N and C-N vibration of the imidazole ring in ZIF-8 crystals [35]. The typical peaks of Dt-COOH(2×) at 1090, 795 and 616 cm^−1^ are ascribed to in-plane Si-O-Si vibration, Si-O deformation and Al-O stretching, respectively [36]. Both characteristic signals of ZIF-8 and Dt-COOH(2×) are observed in the prepared ZIF-8@Dt-COOH(2×). In addition, as shown in Figure 3b, the typical peaks of ZIF-8 at 7.3°, 10.3°, 12.7°, 16.4°, 18.0°, 24.6° and 26.7° and typical peaks of Dt-COOH(2×) at 22° [37] are found in that of ZIF-8@Dt-COOH(2×). These results demonstrate the successful synthesis of ZIF-8 on the surface of Dt-COOH(2×).

### 2.2. Optimization of SPE Conditions

To achieve high recoveries of BZDs with the prepared ZIF-8@Dt-COOH(2×) and enhance the sensitivity of this method, several parameters affecting the SPE performance were investigated, including ionic strength, pH of solution, loading volume, washing solution, elution solvent and elution volume. Eight milliliters of the spiked aqueous solution (10 ng/mL of each BZD) were loaded onto ZIF-8@Dt-COOH(2×) cartridge and all the optimization experiments were repeated three times.

#### 2.2.1. Ionic Strength and Sample pH

Ionic strength and pH of aqueous solution not only affect the molecular states of compounds, but also control the charge species and density of the adsorbent [38]. Consequently, the effect of ionic strength on the recovery was first investigated at an NaCl concentration of 0–15 mmol/L. Figure 4a reveals that when the concentration of NaCl changes from 0 to 15 mmol/L, the recoveries of three BZDs reach 96.6 ± 4.6%–100 ± 3.8%, revealing that ionic strength has no obvious influence on the adsorption of BZDs. In consideration with the common pH range of urine samples (4.8–7.4), the pH of sample solution was also evaluated with pH 3.8–5.6 (0.1 mmol/L HAc-NaAc) and pH 5.8–7.6 (0.1 mmol/L Na_2_HPO_4_-NaH_2_PO_4_). The results are shown in Figure 4b. No significant change in the recoveries of BZDs (92.5 ± 5.7%–101 ± 3.2%) are observed in the investigated pH range of 3.8–7.6. In fact, ultrapure water was also explored, and satisfactory extraction was achieved. Thus, ultrapure water was used to dilute the crude urine sample without any adjustment of pH. These results demonstrate that the electrostatic interaction is not the dominant adsorption interaction between ZIF-8@Dt-COOH(2×) and BZDs.

#### 2.2.2. Loading Volume

The loading volume determines the enrichment capacity and sensitivity of the method. The loading volume from 6 mL to 12 mL was investigated by spiking the constant amount of BZDs (2 μg of each analyte) in aqueous solutions. As shown in Figure 4c, the recoveries of BZDs from 98.4 ± 3.3% to 104 ± 2.3% are obtained with sample volume change from 6 mL to 8 mL. However, further increase in the volume from 10 mL to 12 mL leads to a decrease in adsorption efficiencies to 77.6 ± 4.3%, which are possibly attributed to insufficient contact between BZDs and the adsorbent in large volume. Consequently, the subsequent sample loading volume was carried out with 8 mL.

In addition, the washing process and elution conditions were optimized in detail. The corresponding results and figures are given in the Appendix A. The optimal conditions were shown as follows: (a) washing, NaH_2_PO_4_ (25 mmol/L, pH = 5); (b) elution, 4 mL of MeOH.

### 2.3. Adsorption Performance of ZIF-8@Dt-COOH(2×) for BZDs

A comparison with the bare Dt-COOH(2×) and commercial sorbents (PLS, CX and C_18_) was performed under the same conditions (*n* = 3) to validate the adsorption performance of ZIF-8@Dt-COOH(2×). One hundred fifty milligrams of the adsorbents were packed into the 6 mL empty syringe individually. Subsequently, the diluted urine sample (urine/H_2_O, 3/1, *v*/*v*) (8 mL) was loaded onto the cartridges. Figure 5 illustrates that the adsorption efficiency of BZDs on Dt-COOH(2×) ranges from 20.1% to 39.0%, while ZIF-8@Dt-COOH(2×) provides the significantly enhanced adsorption efficiency of 100%. The results indicate that the presence of ZIF-8 nanoparticles provides the dominant adsorption for the analytes. In addition, ZIF-8@Dt-COOH(2×) has superior or comparable adsorption efficiency with the commercial sorbents PLS (100.0%), CX (94.5–100.0%), and C_18_ (97.6–100%).

Log *P* (octanol–water partition coefficient) refers to the ratio of the solubility of a compound in octanol/water to assess the hydrophobicity of a compound. The higher log *P* value, the stronger the hydrophobicity of the compound. The enhancement factor (EF) of all BZDs was calculated by comparing the extracted analytes concentration with that in the original urine sample. As shown in Table 1, DZP with the highest log *P* value (4.02) offers an EF of 30.0, while TRI with the lowest log *P* value (3.67) gives an EF of 27.8, indicating that the hydrophobic interaction between ZIF-8@Dt-COOH(2×) and BZDs play the key role during the adsorption process given by the hydrophobic characteristics of ZIF-8 [39]. Additionally, the π–π interaction between the aromatic rings of the BZDs and the ZIF-8 framework is another probable reason for the adsorption.

### 2.4. Method Evaluation

Under the optimal conditions, the developed method was investigated in terms of linearity, correlation coefficients (*r*), limit of detection (LOD), limit of quantification (LOQ), accuracy and precision. The standard calibration curves were constructed with standard solutions at 2, 10, 25, 50, 100, 250 and 500 ng/mL (*n* = 3). Table 2 shows that the method exhibits linearity ranging from 2–500 ng/mL with correlation coefficients (*r* ≥ 0.9995) for three BZDs. LODs and LOQs of all analytes in spiked urine samples vary in the range of 0.3–0.4 ng/mL and 1.0–1.3 ng/mL, respectively. Considering that the excretion concentration of BZDs in urine after clinical administration is approximately 10 ng/mL, the recoveries of BZDs at 5, 10 and 20 ng/mL concentrations in blank urine samples were used to assess the accuracy of this method and the average recoveries of three BZDs varied from 80.0% to 98.7%. The intra-day and inter-day precisions (relative standard deviations, RSDs) for three replicate extractions of BZDs were 1.4–5.2% and 1.5–8.2%, respectively. The EFs of all BZDs ranged from 24.0 to 29.6 when the loading volume of the diluted urine sample (urine/H_2_O, 3/1, *v*/*v*) was fixed at 8 mL.

The column-to-column and batch-to-batch precisions were used to evaluate the reproducibility of ZIF-8@Dt-COOH(2×) cartridges (Table 2). The diluted urine sample (urine/H_2_O, 3/1, *v*/*v*) (8 mL) spiked with 20 ng/mL of each analyte was loaded onto ZIF-8@Dt-COOH(2×) cartridges and the RSDs of the recoveries of BZDs were calculated. The column-to-column RSDs of the three ZIF-8@Dt-COOH(2×) cartridges from one batch were 3.3–6.4%, and the batch-to-batch RSDs of three parallel batches ranged from 6.9% to 9.4%, indicating good reproducibility of ZIF-8@Dt-COOH(2×).

### 2.5. Application to Urine Samples

Figure 6 shows the representative HPLC chromatograms of the standard solution (300 ng/mL of each analyte) and urine spiked with 10 ng/mL of each analyte via direct analysis, extracted by ZIF-8@Dt-COOH(2×) and commercial CX adsorbents. Compared with the results of direct analysis (Figure 6b), all BZDs were effectively enriched from complex urine samples by ZIF-8@Dt-COOH(2×) with EFs of 24.0–29.6 and the impurity intensities were significantly decreased (Figure 6c). Furthermore, the recovery (%) was evaluated by the following equation:(1)$$\mathrm{Recovery}\;\left(\%\right)=\frac{C_{found}-C_{real}}{C_{added}}\times 100$$
where C*_added_* is the concentration of the spiked amount of standard. C*_real_* and C*_found_* are the detected concentrations of the analyte before and after the addition of standard in the urine samples, respectively. The recoveries of BZDs obtained by ZIF-8@Dt-COOH(2×) cartridges and commercial CX adsorbents were all higher than 80.0% (Figure 6c,d). These results reveal that the developed technique is effective for the purification and preconcentration of BZDs in urine.

Table 3 shows a comparison of the proposed method with other techniques for the determination of BZDs in biological fluids. The sensitivity of the present method (0.3–0.4 ng/mL) is higher than that using an analogous detector, even a mass spectrometry detector (MS) [10,40,41,42,43,44,45]. Likewise, it also gave lower LODs and comparable recoveries for BZDs than the commercial SPE sorbents of HLB and SLW.

The feasibility of the proposed method was further demonstrated by the analysis of urine samples obtained from the patients taking MID. As listed in Table 4, the concentrations of MID detected in all urine samples are between 10.9 ± 0.7 and 21.7 ± 1.8 ng/mL. The recoveries obtained by spiking 20 ng/mL MID in urine samples range from 81.1% to 109% with RSDs between 3.8% and 9.9%. These results indicate the availability of the proposed method for the determination of BZDs in urine.

## 3. Materials and Methods

### 3.1. Chemicals and Standards

All reagents used were of analytical grade. Diatomite (Dt, 99.95%, mean particle size ~30.8 μm), 3-aminopropyltriethoxysilane (APTES), 2-methylimidazole (2-MeIM), zinc nitrate hexahydrate Zn(NO_3_)_2_·6H_2_O and glutaric anhydride were provided by Aladdin (Shanghai, China). Absolute ethanol was from Tianjin Huihang Chemical Technology Co., Ltd. (Tianjin, China). *N*,*N*′-dimethylformamide (DMF), acetone and ammonium acetate (NH_4_OAc) were acquired from Tianjin Damao Chemical Reagent Factory (Tianjin, China). Acetic acid (HAc), sodium acetate (NaAc), disodium hydrogen phosphate (Na_2_HPO_4_·12H_2_O), sodium dihydrogen phosphate (NaH_2_PO_4_·2H_2_O) and sodium chloride (NaCl) were obtained from Tianjin Zhiyuan Chemical Reagent Co., Ltd. (Tianjin, China). Chromatographic-grade methanol (MeOH) and acetonitrile (ACN) were obtained from Fisher Scientific (Geel, Belgium). Ultrapure water was supplied by Hangzhou Wahaha Group Co., Ltd. (Hangzhou, China). Commercial SPE cartridges, containing hydrophilic–lipophilic balance (PLS), cationic-exchange (CX) and C_18_ were supplied by Dikma (Shanghai, China). The empty polyethylene SPE cartridges (6 mL) were provided by Shanghai Baitaiqi Trading Co., Ltd. (Shanghai, China).

Three benzodiazepine standards containing triazolam (TRI), midazolam (MID) and diazepam (DZP) were from Cerilliant Corp. (Austin, TX, USA). Each stock solution (1.0 mg/mL) was separately prepared in MeOH. Working solutions were obtained by freshly diluting the corresponding stock solutions with MeOH.

### 3.2. Instrumentation

An Agilent 1260 HPLC-DAD (Agilent Technologies, CA, USA) with G1311B pump system, G1329A auto-sampler, G1316A temperature control center and G1315D DAD detector were used for chromatographic analysis. Scanning electron microscope (SEM) micrographs of the prepared materials were provided by a Hitachi S-4800 field emission scanning electron microscope (Hitachi, Tokyo, Japan). Fourier transform infrared (FT-IR) spectra (4000–400 cm^−1^) were achieved on Shimadzu FT-IR-8400S transform infrared spectrometer (Jasco, Kyoto, Japan). X-ray diffraction (XRD) patterns were acquired by a Brucker D8 Venture single crystal X-ray diffractometer (Bruker, Karlsruhe, Germany).

### 3.3. Fabrication of ZIF-8@Dt-COOH

Figure 7 shows the in situ fabrication process of ZIF-8@Dt-COOH. Carboxylate-terminated Dt particles (Dt-COOH) were used as the support material and prepared according to An et al. [46]. Typically, 1.68 g of glutaric anhydride was dispersed in 120 mL of DMF containing APTES (3.48 mL), and the solution was stirred for 3 h in a water bath at 30 °C. Subsequently, the dispersion solution of Dt (2.0 g) in the mixture of DMF (100 mL) and H_2_O (9 mL) was added to the above solution. The resultant solution was mechanically stirred for another 5 h at 30 °C. The collected Dt-COOH particles were washed with ultrapure water and absolute ethanol consecutively, and then dried in vacuum overnight at 25 °C. The obtained materials were named as Dt-COOH(1×) and Dt-COOH(2×), respectively.

For the growth of ZIF-8 nanocrystals onto Dt-COOH particles, Dt-COOH (1.0 g) was added to MeOH (60 mL) solution containing 2.86 g of Zn(NO_3_)_2_·6H_2_O and stirred for 5 min in a 70 °C water bath. During this process, the affinity of Zn^2+^ for the carboxyl group results in the adsorption of Zn^2+^ on the surface of Dt-COOH and the obtained Zn^2+^@Dt-COOH was used as a precursor for the next reaction. Subsequently, 60 mL of MeOH solution containing 7.88 g 2-MeIM was added, and the reaction was maintained for further 15 min with mechanical stirring at 70 °C. The obtained ZIF-8@Dt-COOH was washed with ultrapure water and absolute ethanol consecutively. Finally, the composite was dried at 60 °C under vacuum for 3 h. Parameters of Zn^2+^ concentration (0.04, 0.08 and 0.12 mmol/mL) and carboxylation cycle of Dt (1 and 2) were studied. ZIF-8 nanocrystals were also prepared for comparison.

### 3.4. Sample Pretreatment

This work was approved by the Ethics Committee of North China University of Science and Technology (Tangshan, China). Urine samples were provided by volunteers who were administered MID from North China University of Science and Technology Affiliated Hospital (Tangshan, China). Blank urine samples from healthy volunteers (drug-free) from North China University of Science and Technology were used as controls, and all samples were stored in a refrigerator at 4 °C. The corresponding BZD standards were added to the crude urine fluids to obtain spiked urine samples. Eight milliliters of the diluted urine sample (urine/H_2_O, 3/1, *v*/*v*) was centrifuged at 1500 rpm for 10 min, and the supernatant was selected for analysis.

### 3.5. SPE Procedure

To prepare SPE cartridges, ZIF-8@Dt-COOH(2×) adsorbents (150 mg) were packed in the 6 mL empty syringe and lugged with polypropylene disks to prevent the loss of adsorbent particles. After being preconditioned with 2 mL of MeOH and ultrapure water, the cartridge was available for sample pretreatment. The diluted urine sample (urine/H_2_O, 3/1, *v*/*v*) (8 mL) was loaded onto the cartridge with a flow rate of 0.4 mL/min. Then, the cartridge was washed with 4 mL of NaH_2_PO_4_ solution (25 mmol/L, pH = 5) and eluted with 4 mL of MeOH. The collected eluent was evaporated to dryness under N_2_ gas stream at 35 °C. Finally, the residues were reconstituted with a 0.2 mL mixture of MeOH and H_2_O (65/35, *v*/*v*), and filtered using a 0.22 μm filter membrane for HPLC analysis.

Commercial PLS, CX and C_18_ adsorbents were also employed for the pretreatment of the urine in the optimized conditions for comparison.

### 3.6. HPLC Analysis

HPLC analysis was performed on a C_18_ column (250 mm × 4.6 mm, 5 μm, Agilent, CA, USA). The mixture of MeOH and H_2_O was performed as a mobile phase at a flow rate of 1 mL/min. The gradient elution procedure was set as 0-9 min, 62% to 70% MeOH, 9–12 min, 70% MeOH. The detection wavelength was 228 nm, and the injection volume was 20 μL.

## 4. Conclusions

In this work, a novel ZIF-8@Dt-COOH composite was synthesized using carboxylate-terminated Dt particles as nucleation, Zn(II) as the central ion and 2-methylimidazole as the ligand by controllable in situ reaction within 20 min. The prepared ZIF-8@Dt-COOH was developed as an SPE adsorbent for isolation of three BZDs in urine, and, compared with the bare Dt-COOH, the extraction efficiency for BZDs was significantly improved from 20.1–39.0% to 100%. In addition, it exhibited superior extraction performance in comparison with commercial PLS, CX and C_18_ sorbents. The interaction of π–π and hydrophobicity between BZDs and ZIF-8 are responsible for the high adsorption of BZDs on ZIF-8@Dt-COOH with EFs of 24.0–29.6. These results confirm that the proposed method could be utilized in the preconcentration of BZDs in urine samples.

## Figures and Tables

**Figure 1 molecules-26-05209-f001:**
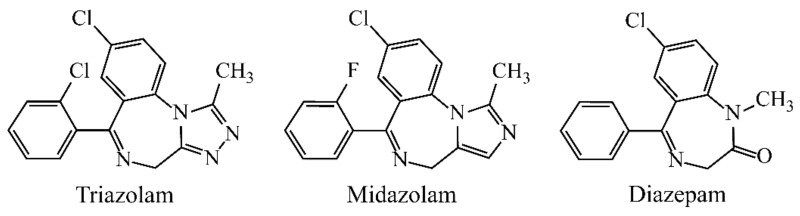
Chemical structures of three BZDs.

**Figure 2 molecules-26-05209-f002:**
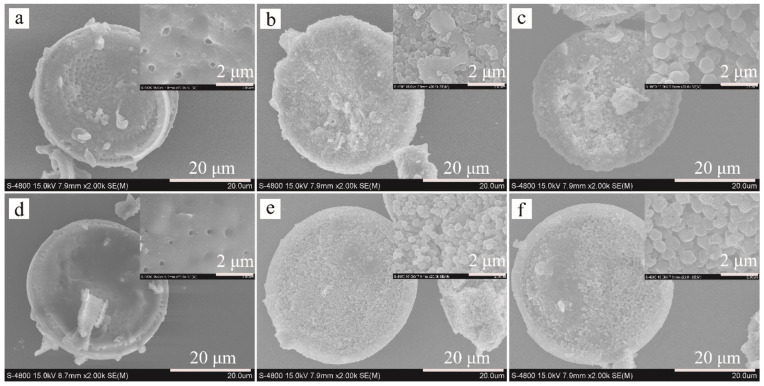
SEM images of (**a**–**c**) ZIF-8@Dt-COOH(1×) and (**d**–**f**) ZIF-8@Dt-COOH(2×) with Zn^2+^ concentrations of 0.04, 0.08 and 0.12 mmol/mL (2000× and inlets, 20,000×).

**Figure 3 molecules-26-05209-f003:**
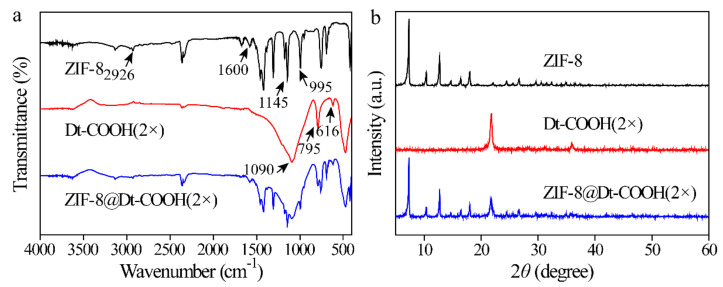
(**a**,**b**) FT-IR spectra and XRD patterns of ZIF-8, Dt-COOH(2×) and ZIF-8@Dt-COOH(2×).

**Figure 4 molecules-26-05209-f004:**
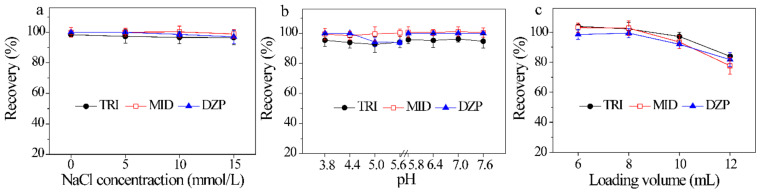
Effects of (**a–c**) ionic strength of sample solution, pH of sample solution and loading volume on the recoveries of three BZDs (*n* = 3).

**Figure 5 molecules-26-05209-f005:**
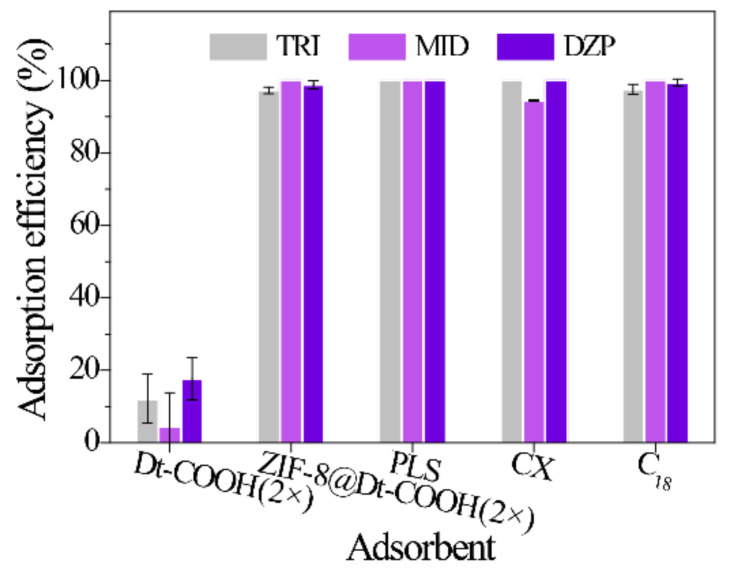
Effect of different SPE sorbents on the adsorption efficiency of three BZDs (*n* = 3).

**Figure 6 molecules-26-05209-f006:**
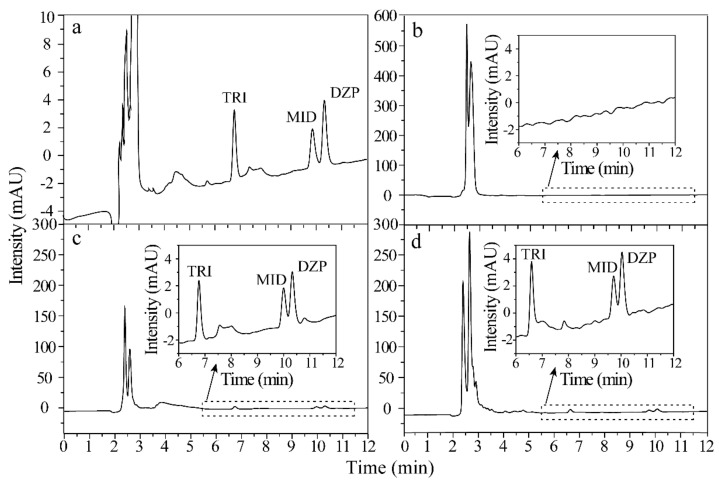
Chromatograms of (**a–d**) BZDs in the standard solution (300 ng/mL of each analyte), spiked urine sample (10 ng/mL of each analyte) with direct injection, pretreated by ZIF-8@Dt-COOH(2×) SPE cartridge and pretreated by commercial CX SPE adsorbents.

**Figure 7 molecules-26-05209-f007:**
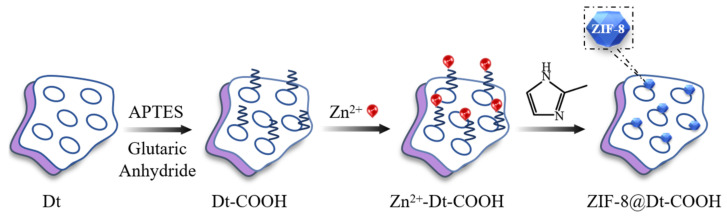
The fabrication flow-chart of ZIF-8@Dt-COOH.

**Table 1 molecules-26-05209-t001:** Log *P* and p*Ka* of the analytes and EF, adsorption efficiency of the proposed materials for BZDs.

Analyte	Log *P* ^a^	p*Ka* ^b^	EF	Adsorption Efficiency (%)	
				Dt-COOH(2×)	ZIF-8@Dt-COOH(2×)
TRI	3.67	1.5	27.8	20.1	100
MID	3.76	1.7	30.0	39.0	100
DZP	4.02	3.3	30.0	36.1	100

^a^ Log *P* was defined as the ratio of octanol/water partition; ^b^ p*Ka* was defined as the dissociation constant.

**Table 2 molecules-26-05209-t002:** Linear range, regression equation (*r*), LOD, LOQ, recovery, precisions and reproducibility of the proposed method.

Analyte	Linear Range (ng/mL)	Regression Equation ^a^ (*r*)	LOD (ng/mL)	LOQ (ng/mL)	Spiked (ng/mL)	Recovery (%)	Precisions (RSD, %, *n* = 3)		Reproducibility ^b^	
							Intra-Day	Inter-Day	Column-to-Column	Batch-to-Batch
TRI	2–500	*y* = 1.1317 − 5.615 (0.9995)	0.3	1.0	5	92.9	5.2	5.7	3.3	6.9
					10	86.8	1.4	2.5		
					20	96.7	1.5	1.5		
MID	2–500	*y* = 0.8581*x* − 3.349 (0.9996)	0.4	1.3	5	98.7	4.1	7.3	6.4	8.0
					10	83.9	3.7	3.1		
					20	81.3	1.9	4.3		
DZP	2-500	*y* = 1.3859*x* − 6.372 (0.9996)	0.3	1.0	5	98.0	2.9	7.5	3.8	9.4
					10	80.0	2.6	2.9		
					20	86.5	1.9	8.2		

^a^*y*, peak area; *x*, mass concentration, ng/mL; ^b^ the reproducibility was assessed by calculating the column-to-column and batch-to-batch RSDs (*n* = 3) of BZDs.

**Table 3 molecules-26-05209-t003:** Comparison of the proposed method with other methodologies for the determination of BZDs in biological fluids.

Pretreatment Method	Instrument	EF ^a^	LOD (ng/mL)	Recovery (%)	References
LLE	LC-MS/MS	~10	0.01–0.5	81–95	[40]
SPDE (HLB) ^b^	LC-TOF-MS	~10	10	89.6–105.0	[41]
SPE (SLW)	UPLC-MS/MS ^c^		0.3	65.3–114.3	[10]
MAE ^d^	HPLC	5	6.2–12.6	89.8–102.1	[42]
SPE (HLB)	HPLC	5	3	68.5–97.6	[43]
DNUM ^e^	HPLC	23.1–24.0	1.2–1.5	92.2–96.0	[44]
D-μ-SPE	HPLC	27.7–32.8	0.2–2.0	84.0–99.0	[45]
SPE (ZIF-8@Dt-COOH)	HPLC	24.0–29.6	0.3–0.4	80.0–98.7	This work

^a^ Enhancement factor, ^b^ hydrophilic–lipophilic balanced copolymer, ^c^ ultra-performance liquid chromatography-tandem mass spectrometry, ^d^ microwave-assisted extraction, ^e^ dispersive nanomaterial ultrasound-assisted microextraction.

**Table 4 molecules-26-05209-t004:** Analytical results of MID in urine samples from the patients administered MID.

Sample	Found ± SD (ng/mL)	Recovery ^a^ (%)	RSD (%)
1	12.4 ± 1.2	81.1	9.9
2	10.9 ± 0.7	94.9	6.4
3	14.8 ± 1.2	91.9	8.0
4	15.1 ± 1.3	101	8.3
5	21.4 ± 0.7	109	3.8
6	21.7 ± 1.8	100	7.0

^a^ The spiked level was 20 ng/mL of MID (*n* = 3).

## Data Availability

The data presented in this study are available on request from the corresponding author.

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
