# Peer review of "Facile Fabrication of Diatomite-Supported ZIF-8 Composite for Solid-Phase Extraction of Benzodiazepines in Urine Samples Prior to High-Performance Liquid Chromatography"

_molecules, 2021, doi:10.3390/molecules26175209_

Round 1

Reviewer 1 Report

The manuscript reports on the synthesis of a diatomite supported zeolitic imidazole framework-8 sorbent and its application as a sorbent for the extraction of benzodiazepines from urine samples.

The paper provides a comprehensive account of the study and the work is interesting in that shows the suitability of the proposed methodology for the determination of the target analytes. The work is well organized and covers sufficiently its aims. It includes a systematic evaluation on the relevant parameters affecting method performance. The developed methodology is successfully compared with other SPE methods reported in the literature. The proposed method uses a fairly simple sorbent preparation procedure and gives satisfactory precision and limits of detections for the selected benzodiazepines.

In general, I believe this manuscript merits publication, and have only a few comments and suggestions to the authors:

  • Recently, the authors have published several papers in reporting the synthesis and application of similar sorbents in dispersive solid phase extraction mode (Microchimica Acta (2020)187(9) 54, Microchemical Journal (2020) 157 105062, Analytical Methods (2020) 12(31) 3924-3932). I think these studies should be referenced and commented in the manuscript. Moreover, the comparison of the performance of the ZIF-8@Dt-COOH sorbent in Table 3, should include other studies using MOFs in dispersive solid phase extraction of benzodiazepines in urine (i.e.,  Journal of Chromatography B: Analytical Technologies in the Biomedical and Life Sciences (2016) 1008 146-155, Analytica Chimica Acta (2014) 844 80-89).
  • The amount of sorbet used throughout this study was 150 mg, was this parameter optimized?
  • The effect of the pH was studied in the range 3.8-7.6 and no influence on the recovery of the analytes was observed. Which pH was finally selected for the application to real samples? I could not find this information in the manuscript.
  • In my opinion Table S1 (in supplementary materials) adds little value. It could be eliminated since the relevant information is already included in the text.

Author Response

请看附件

Reviewer 2 Report

The work is interesting. However, a number of parts in manuscript are not clearly nor adequately described. e.g., 

  1. in figure 6, for (b), direct analysis, what are the details? Is it the blank urine sample after direct injection? If it is direct injection, how the analytes be detected without concentration?
  2. How is the recovery % obtained/calculated? It is given that recovery is obtained from the spike urine sample. However, the exact procedure is not given.
  3. How is the enhancement factor obtained from the original urine sample concentration and extracted analytes concentration? 
  4. Why there are no enhancement factors for LLE, SPDE (HLB) and SPE (SLW)?
  5. Superscript "c" for SPE (HLB) is UPLC-tandem mass spectrometry, but the instrument indicated is HPLC only.

Author Response

请看附件
